# Cilostazol Attenuates Hepatic Steatosis and Intestinal Disorders in Nonalcoholic Fatty Liver Disease

**DOI:** 10.3390/ijms25116280

**Published:** 2024-06-06

**Authors:** Tianqi Min, Shuting Qiu, Yan Bai, Hua Cao, Jiao Guo, Zhengquan Su

**Affiliations:** 1Guangdong Engineering Research Center of Natural Products and New Drugs, Guangdong Provincial University Engineering Technology Research Center of Natural Products and Drugs, Guangdong Pharmaceutical University, Guangzhou 510006, China; 15618039319@163.com (T.M.); q743160056@163.com (S.Q.); 2Guangdong Metabolic Disease Research Center of Integrated Chinese and Western Medicine, Key Laboratory of Glucolipid Metabolic Disorder, Ministry of Education of China, Guangdong TCM Key Laboratory for Metabolic Diseases, Guangdong Pharmaceutical University, Guangzhou 510006, China; 3School of Public Health, Guangdong Pharmaceutical University, Guangzhou 510310, China; angell_bai@163.com; 4School of Chemistry and Chemical Engineering, Guangdong Pharmaceutical University, Zhongshan 528458, China; caohua@gdpu.edu.cn

**Keywords:** nonalcoholic fatty liver disease, cilostazol, liver lipids, intestinal flora, gluconeogenesis

## Abstract

Nonalcoholic fatty liver disease (NAFLD) is one of the most common chronic liver diseases in the world, which begins with liver lipid accumulation and is associated with metabolic syndrome. Also, the name chosen to replace NAFLD was metabolic dysfunction-associated steatotic liver disease (MASLD). We performed focused drug screening and found that Cilostazol effectively ameliorated hepatic steatosis and might offer potential for NAFLD treatment. Our aim was to investigate the therapeutic effects of Cilostazol on the glycolipid metabolism and intestinal flora in NAFLD mice and explore the specific mechanism. In this study, 7-week-old male C57BL/6J mice were fed a high-fat diet (HFD) for 8 weeks to induce NAFLD, and then treated with intragastric administration for 12 weeks. The results showed that Cilostazol inhibited liver lipid de novo synthesis by regulating the AMPK-ACC1/SCD1 pathway and inhibited liver gluconeogenesis by the AMPK-PGC1α-G6P/PEPCK pathway. Cilostazol improved the intestinal flora diversity and intestinal microbial composition in the NAFLD mice, and specifically regulated Desulfovibrio and Akkermansia. In addition, Cilostazol increased the level of short-chain fatty acids in the NAFLD mice to a level similar to that in the blank Control group. Cilostazol reduces liver lipid accumulation in NAFLD mice by improving glucose and lipid metabolism disorders and intestinal dysfunction, thereby achieving the purpose of treating NAFLD.

## 1. Introduction

Over the last 40 years, the global prevalence of nonalcoholic fatty liver disease (NAFLD) has been 30.05%. However, the vast majority of patients with NAFLD are asymptomatic until advanced and potentially irreversible liver damage occurs [1]. NAFLD is often associated with obesity, diabetes, hypertension, and hyperlipidemia [2]. Since it is associated with these comorbidities, it was renamed MASLD [3]. Atherosclerotic cardiovascular disease is the main cause of death in patients with NAFLD [4]. In addition, malignant tumors are one of the main causes of death from NAFLD, which are mainly driven by extrahepatic malignant tumors, followed by hepatocellular carcinoma [5,6]. NAFLD is a major global economic and health care burden [7]. The Multiple-Hit hypothesis considers multiple insults acting together on genetically predisposed subjects to induce NAFLD and provides a more accurate explanation of NAFLD pathogenesis. Such hits include insulin resistance, hormones secreted from the adipose tissue, nutritional factors, the gut microbiota, and genetic and epigenetic factors. In the Multiple-Hit pathogenesis, the pathophysiological changes in NAFLD are induced by many factors [8]. In NAFLD, the lipids initially accumulate in hepatocytes and develop into reversible hepatic steatosis, followed by the infiltration of immune cells in the liver, consequently hastening the inflammatory process and triggering liver fibrosis [9]. 

An imbalance between liver lipid storage and lipid clearance, such as increased de novo lipogenesis and fatty acid content, can lead to liver cell damage and lead to peripheral insulin resistance, further promoting lipid production, with glucose and fatty acids as the substrates [10]. Gluconeogenesis is the main way of endogenous glucose production in the body during fasting or starvation. In the process of pathological overnutrition, insulin resistance can promote liver gluconeogenesis, with glycerol and pyruvate as the substrates, thus promoting liver lipid synthesis and accelerating the progress of NAFLD [11]. Therefore, regulating liver lipid production and gluconeogenesis to avoid lipid accumulation is a significant therapeutic goal for NAFLD.

Some studies have shown that there is a link between intestinal flora abnormalities, intestinal barrier damage, liver inflammation, and metabolic abnormalities under high-fat-diet conditions [12,13]. The imbalance of intestinal microflora leads to the enhancement of the bacterial penetration ability, an decrease in the overall thickness of the mucus layer, and the redistribution of epithelial barrier tight junction proteins [14]. When the translocation of microorganisms and their metabolites increases, LPS, which can cause endotoxemia in the blood, can exacerbate liver inflammation [15]. Consistent with the preclinical model, the incidence of intestinal bacterial overgrowth and changes in the microbial composition in patients with NAFLD were higher than those in healthy controls [16,17]. Indigestible carbohydrates provided by the diet (such as dietary fiber) can form short-chain fatty acids (SCFAs) under bacterial fermentation [18]. SCFAs can further reduce liver steatosis and insulin sensitivity by reducing the expression of PPARγ [19]. In consequence, intestinal interventions are the future trend in the treatment of chronic liver disease.

Cilostazol is a 2-hydroxyquinolone derivative and a drug approved for improving the claudication distance [20]. Other studies have shown that Cilostazol can ameliorate hepatic steatosis, but the specific mechanism is still unknown [21]. We performed focused drug screening and found that Cilostazol could be a potential drug to treat hyperlipidemia-related diseases such as NAFLD. Cilostazol has been reversibly inhibiting phosphodiesterase III through its active metabolite for many years, thereby reducing the degradation of cyclic adenosine monophosphate and inhibiting platelet reactivity and aggregation [22]. Some studies have shown that Cilostazol treatment to prevent secondary stroke is associated with a decreased quantity of triglycerides and increased high-density lipoprotein cholesterol [23]. In terms of lipid metabolism, the effect of Cilostazol on liver adipogenesis is related to the inhibition of SREBP-1c expression and the enhancement of STAMP2 expression through AMPK [24,25]. Liver lipid degeneration is closely related to the occurrence of NAFLD.

On the basis of these data, we examined whether Cilostazol ameliorates lipid accumulation in a high-fat-diet-induced fatty liver animal model. We investigated the mechanism of Cilostazol in the treatment of NAFLD, which was studied based on the AMPK pathway, and the effect of Cilostazol on the intestine of NAFLD mice was ulteriorly discussed. This study can effectively provide new insights into research and the application evaluation of Cilostazol and offer a novel therapeutic option for NAFLD focused on targeting intermediary metabolism.

## 2. Results

### 2.1. Cilostazol Reduced Body Weight and Liver Weight in NAFLD Mice

In Figure 1A, compared with the Control group, the average food intake of the mice in the Model group, Metformin group, and Cilostazol group was lower, which was speculated to be related to the softness of the high-fat diet and the poor palatability of the normal diet. Compared with the Model group, Metformin and Cilostazol had no appetite inhibition effect on the NAFLD mice and significantly reduced the body weight of the NAFLD mice. At the same time, it can be seen from Figure 1A that compared with the Control group, the liver weight and liver index of the Model group increased significantly, indicating that an HFD can cause liver hypertrophy in NAFLD mice. Cilostazol intervention significantly reduced the liver weight and liver index of the mice and caused liver hypertrophy in the NAFLD mice.

### 2.2. Cilostazol Improved Liver Function and Lipid Accumulation in NAFLD Mice

In order to explore the effect of the Cilostazol sample on the liver function in mice, ALT and AST were detected in each group. The results are shown in Figure 1B. The levels of AST and ALT in the sera of middle- and high-dose Cilostazol mice were significantly lower than those in the Model group and close to those in the Control group. The results of HE staining are shown in Figure 1B. The liver cells of the Control group were arranged regularly, and the cytoplasm was uniform. There was a large number of lipid droplets with different sizes in the cytoplasm of the liver cells of the Model group, and the cell boundaries were blurred. The fat vacuoles in the Cilostazol group were significantly reduced compared with those of the Model group, and the arrangement of hepatocytes was significantly improved. The results of oil red O staining are shown in Figure 1B. Compared with the Control group, the liver of the Model group showed a large number of red lipid droplets, and the content of lipid droplets in the liver of the Cilostazol group decreased. Cilostazol can reduce liver steatosis and improve liver function in NAFLD mice.

### 2.3. Cilostazol Alleviated Dyslipidemia and Insulin Resistance in NAFLD Mice

In order to explore the effect of Cilostazol on the serum lipid levels in mice, the levels of TG, TC, LDL-C, HDL-C, and FFA in each group were detected. The results are shown in Figure 2A. Compared with the Control group, the detection indexes of the Model group were significantly abnormal, causing hyperlipidemia. Compared with the Model group, the contents of TG, TC, LDL-C, and FFA in the Metformin and Cilostazol groups were significantly decreased, while the content of HDL-C was increased. In order to explore the IR of mice, the fasting blood glucose (FBG) level, fasting insulin level, and HOMA-IR index were measured and analyzed. The results are shown in Figure 2A. The fasting blood glucose level and HOMA-IR index in the Metformin and Cilostazol groups were significantly lower than those in the Model group, but close to those in the Control group, indicating that Cilostazol had a certain hypoglycemic effect and alleviated glucose tolerance in the mice.

### 2.4. Cilostazol Alleviated Systemic Inflammation and Liver Lipid Levels in NAFLD Mice

In order to explore the effect of the Cilostazol samples on systemic inflammation in the mice, IL-6, IL-1β, IL-10, and TNF-α were detected in each group. The results are shown in Figure 2B. Compared with the Control group, the content of IL-1β in the Model group was significantly increased, and the content of IL-10 was significantly decreased. Cilostazol can reduce the level of pro-inflammatory factors in mice, increase the level of anti-inflammatory factors, and effectively cause inflammation in mice. In order to explore the effect of Cilostazol on the liver lipid levels in the mice, liver TG, TC, LDL-C, and HDL-C were detected in each group of mice. The results are shown in Figure 2B. Compared with the Control group, the contents of TG, TC, and LDL-C in the Model group increased significantly. Cilostazol significantly decreased liver TG levels in the NAFLD mice, and also had a tendency to improve liver TC, LDL-C, and HDL-C in the mice.

### 2.5. Cilostazol Inhibits De Novo Lipogenesis (DNL) in the Liver of NAFLD Mice by Activating AMPK-ACC1/SCD1 Pathway

Increased liver DNL in NAFLD is an important factor in liver lipid elevation. Firstly, the relative quantitative analysis of gene expression related to lipid synthesis was performed by RT-PCR. The results are shown in Figure 3A. The expression levels of SREBP-1c, ACC1, SCD1, FAS, ACC2, and HMGCR mRNA in the liver of the Model group were significantly higher than those in the Control group. Compared with the Model group, the expression of these genes in the Cilostazol medium-dose group and the high-dose group was significantly reduced. In addition, CPT1A mRNA expression was significantly increased in the Cilostazol-H group. The expression of AMPK, ACC1, and SCD1 proteins was further analyzed, and the results are shown in Figure 3B. Compared with the Model group, high-dose Cilostazol significantly increased the phosphorylation of AMPK and ACC1 in mice, activated AMPK, and decreased the expression of ACC1 and SCD1 proteins. Cilostazol can inhibit liver DNL by activating the AMPK-ACC1/SCD1 pathway.

### 2.6. Cilostazol Reduced Hepatic Gluconeogenesis in NAFLD Mice by Inhibiting AMPK-PGC1α-G6P/PEPCK Pathway

When the metabolites produced in the case of pathological overnutrition continue to promote gluconeogenesis, excessive glucose can also cause liver lipid accumulation. Firstly, the gene and protein expression levels of PGC1α, PEPCK, and G6P in the mouse livers were analyzed. The results are shown in Figure 4. The expression levels of PGC1α, PEPCK, and G6P in the liver of the Model group were significantly higher than those of the Control group. After Cilostazol-H intervention, these genes and proteins were significantly reduced. In addition, the quantitative analysis of AMPK in the mouse livers showed that the ratio of p-AMPK/AMPK was significantly higher than that of the Model group after Cilostazol intervention, and the phosphorylation level of AMPK was significantly increased. Cilostazol can promote the phosphorylation of AMPK, inhibit the expression of downstream PGC1α, and reduce the gene and protein expression of G6P and PEPCK to inhibit the gluconeogenesis process in mouse livers.

### 2.7. Cilostazol Improved the Intestinal Barrier of NAFLD Mice

LPS in the blood can cause endotoxemia, and the tight junction composed of transmembrane proteins, such as Occludin and Claudin-1, is disrupted, resulting in increased intestinal permeability and subsequent liver inflammation. In this study, the level of LPS in the mouse sera was determined, and the results are shown in Figure 5A. The serum LPS level of the mice in the Model group was significantly higher than that in the Control group, and the serum LPS level of the mice was significantly decreased after Cilostazol intervention. In order to evaluate the effect of Cilostazol on the morphology of the colon tissue in the mice, the HE staining of the colon tissue was performed. The results are shown in Figure 5B. Lymphocyte infiltration was observed in the colon of the Model group, the number of epithelial goblet cells decreased, and the glands were irregularly arranged. Cilostazol group improved this situation. RT-PCR and WB were used to quantify the tight junction proteins in the colon tissue. The results are shown in Figure 5C. Compared with the Model, Cilostazol-H significantly increased the expression of Claudin-1, Occludin, and ZO-1 in the colon. The Cilostazol low-dose group and middle-dose group mice also showed a similar trend.

### 2.8. Cilostazol Affects the Diversity of Intestinal Flora in NAFLD Mice

In order to measure the species diversity of intestinal flora and the differences between bacterial communities, α diversity analysis and β diversity analysis were performed on the intestinal colonies of mice in the Control group, Model group, and Cilostazol group. The results are shown in Figure 6A,B. The ACE, Chao1, and Shannon indexes of the Cilostazol group were higher than those of the Model group, and the Simpson index was lower than that of the Model group. In PCA analysis, the intestinal colony composition of the three groups of mice had significant separation. In PCoA analysis, there was no effective separation between the Cilostazol and Model groups, which was significantly different from the intestinal flora of the Control group. In NMDS analysis, the stress value was 0.164, indicating that the model had a good discrimination effect, that is, the colony structure between the three groups was different. Cilostazol increased the diversity of intestinal flora in the NAFLD mice.

### 2.9. Cilostazol Changed the Flora Structure of NAFLD Mice

The families with a high average abundance of intestinal flora in the three groups of mice at the phylum, family, and genus levels were analyzed. The results are shown in Figure 6C–F. At the phylum level, compared with the Model group, the relative abundance of Verrucomicrobiota, Bacteroidota, and Campylobacterota increased after Cilostazol intervention, and the relative abundance of Proteobacteria, Actinobacteriota, Desulfobacterota, Deferribacteres, and Firmicutes decreased. At the same time, the value of Firmicutes/Bacteroidetes in the Cilostazol group was significantly lower than that in the Model group. At the family level, compared with the Model group, the Cilostazol group significantly increased the proportions of Akkermansiaceae, Oscillospiraceae, and Bacteroidaceae, and significantly decreased the proportions of Lactobacillaceae, Desulfovibrionaceae, and Erysipelatoclostridiaceae. At the genus level, compared with the Control group, the proportions of Desulfovibrio and Blautia in the Model group were significantly increased, and those of Lachnospiraceae_NK4A136_group and Dubosiella were significantly decreased. Compared with the Model group, the Cilostazol group significantly increased the proportions of Akkermansia and Bacteroides and significantly decreased the proportions of Desulfovibrio, Ligilactobacillus, and Blautia. Cilostazol can improve the intestinal flora structure of NAFLD mice.

### 2.10. Analysis of Marker Species of Inter-Group Differences in Intestinal Flora of Mice

LEfSe analysis based on the linear discriminant analysis (LDA) effect size was used to determine the special flora of intestinal microorganisms affected by Cilostazol. The results are shown in Figure 7A. An LDA score greater than the set value of 4.0 was used to screen out the marker species of inter-group differences. The Model group was significantly enriched with g_Blautia, s_Desulfovibrio_fairfieldensis, s_Lachnospiraceae_bacterium_28_4, and s_Faecalibaculum_rodentium. The Cilostazol group was significantly enriched with s_Akkermansia_muciniphila, f_Akkermansiaceae, p_Verrucomicrobiota, c_Verrucomicroblae, o_Verrucomicrobiales, and g_Akkermansia. An evolutionary branch diagram is used to better show the differences between groups from the phylum to genus classification levels, as shown in Figure 7B. The synergy between these core floras led to community differences in each group. The Tax4fun method was used to predict the function of microorganisms in the intestinal cecal contents of the mice. The results are shown in Figure 8. The functional genes of intestinal flora in Model group and Cilostazol group were enriched in carbohydrate metabolism and amino acid metabolism.

### 2.11. Correlation Analysis between Intestinal Bacteria and NAFLD Indexes in Mice

In order to study the relationship between the intestinal flora and NAFLD-related indicators, such as blood lipid indicators and inflammatory factors, a correlation heat map between intestinal flora and NADLD indicators was drawn by using R software (V4.0.0). The results are shown in Figure 7C. Spearman correlation analysis showed that the NAFLD-related indicators (TG, TC, LDL-C, FFA, ALT, AST, IL-6, IL-1β, TNF-α, and LPS) were mainly positively correlated with Desulfovibrio, Lactococcus, Limosilactobacillus, Candidatus_Saccharimonas, Blautia, and other bacteria. They were mainly negatively correlated with Parabacteroides and Lachnospiraceae_NK4A136_group. HDL-C and IL-10 were positively correlated with Akkermansia and Lachnospiraceae_NK4A136_group and negatively correlated with Desulfovibrio, Romboutsia, and Lactococcus. Cilostazol can regulate metabolic disorders in NAFLD mice through intestinal flora, and Desulfovibrio is significantly correlated with NAFLD-related indicators.

### 2.12. Cilostazol Affected the Intestinal Flora Metabolites SCFAs in NAFLD Mice

SCFAs can interfere with liver lipid metabolism by inhibiting DNL and promoting fatty acid β-oxidation. In this study, the levels of SCFAs in the cecal contents of three groups of mice were analyzed by GC-MS, and the results are shown in Figure 9A. The content in the mice in the Cilostazol group was similar to that in the Control group. Compared with the Model group, Cilostazol administration up-regulated the contents of acetic acid, butyric acid, hexanoic acid, and propionic acid in the cecal contents, down-regulated the contents of isobutyric acid, isovaleric acid, and valeric acid, and reversed the trend of SCFAs content in the Model mice. In order to study the relationship between the intestinal flora and its metabolite SCFAs, a correlation heat map between them was drawn by using R software (V4.0.0). The results are shown in Figure 9B. Acetic acid, propionic acid, and butyric acid were positively correlated with Lachnospiraceae_NK4A136_group, Bifidobacterium, and Anaerostipes and negatively correlated with Desulfovibrio, Bilophila, and Blautia. Caproic acid was strongly negatively correlated with [Eubacterium]_fissicatena_group. Valproic acid was positively correlated with Desulfovibrio and negatively correlated with Parabacteroides. Isobutyric acid and isovaleric acid were positively correlated with Anaerotruncus, Blautia, Bilophila, and Colidextribacter and negatively correlated with [Eubacterium]_ruminantium_group, [Eubacterium]_xylanophilum_group, and Lachnospiraceae_NK4A136_group.

## 3. Discussion

Cilostazol is mainly used to treat intermittent claudication, an early symptom of peripheral arterial disease. The intermittent claudication symptoms are an indicator of systemic atherosclerosis, and people with intermittent claudication are three to six times more likely to die of cardiovascular disease than their peers without intermittent claudication [20]. The main pharmacological active metabolite of Cilostazol is 3,4-dehydrocilostazol, which has the effects of reducing serum triglyceride, increasing the quantities of high-density lipoproteins and antithrombotics [26]. The risk of cardiovascular events in patients with NAFLD increased by 65%, including atherosclerosis [27]. Based on the fact that Cilostazol can regulate blood lipids and treat cardiovascular diseases, it is speculated that it also has a beneficial effect on NAFLD, and the effect of Cilostazol on NAFLD has been studied.

The diet of mice has an important impact on body weight. The intake of high-calorie food is more likely to cause lipid production, such as cholesterols and triglycerides, in the body. In this experiment, compared with the Model group, the food intake of the Cilostazol group did not change significantly, but the body weight of the mice decreased significantly at the end of the experiment. More and more evidence shows that weight loss can improve NAFLD [28]. It shows that Cilostazol can improve the condition of NAFLD mice by reducing their body weight.

The initial stage of hepatic steatosis involves the ectopic accumulation of triglycerides in the liver. In an insulin-resistant environment, the excessive hydrolysis of triglycerides in adipose tissue causes an increase in free fatty acids. Most of the triglycerides synthesized in the liver are derived from free fatty acids. The imbalance of lipid metabolism causes the production of lipotoxic substances, which, in turn, causes oxidative stress and endoplasmic reticulum stress, inflammasome activation, and subsequent inflammatory stimulation, leading to the progression of NAFLD [29]. The serum levels of TG, TC, LDL-C, and FFA in the Model group were significantly increased, and the levels of insulin and fasting blood glucose were also significantly increased, and the HOMA-IR index was significantly higher than that in the Control group. At the same time, the levels of serum AST and ALT in the Model group were abnormally increased, suggesting that the disorder of glucose and lipid metabolism in the mice had caused liver damage. Cilostazol effectively improved the abnormal glucose and lipid metabolism and liver function in the NAFLD mice. Cilostazol also effectively alleviated HFD-induced inflammation in the mice, especially in the high-dose group. Cilostazol significantly reduced the accumulation of lipid droplets in the livers of the NAFLD mice. This shows that Cilostazol has a therapeutic effect on NAFLD.

AMPK activation is a valuable therapeutic option for NAFLD, a disease associated with increased fatty acid production. DNL is significantly increased in patients with NAFLD compared with healthy Controls. ACC1 and ACC2 are the first rate-limiting steps in catalyzing DNL [30]. ACC mainly acts on acetyl coenzyme A to form the fatty acid synthesis substrate malonyl coenzyme A to synthesize the fatty acids [31,32]. As an allosteric inhibitor of CPT1, malonyl-CoA can inhibit the expression of CPT1 and inhibit fatty acid oxidation [33]. AMPK activation inhibits ACC phosphorylation to inhibit lipid synthesis and increase CPT1 activity to promote fatty acid oxidation. It can be seen that the AMPK-ACC pathway can be used to treat diseases caused by excessive fatty acids. The transcription factor SREBP-1c in the upstream signaling pathway of ACC is involved in the regulation of ACC and SCD1, leading to liver lipid metabolism disorders. FAS can react with malonyl coenzyme A to form palmitate, which is the first fatty acid product in DNL [34]. In addition, the main pathogenic factor of NAFLD is liver insulin resistance. Insulin in patients with NAFLD cannot inhibit the production of endogenous glucose, while glucose is the main substrate of liver DNL, and insulin resistance can lead to more free fatty acid release. Hepatic gluconeogenesis is the main source of endogenous glucose. Insulin resistance cannot effectively inhibit hepatic gluconeogenesis, thereby aggravating DNL in the liver. As a transcription element of insulin signal transduction, PGC1α is a key regulator of glucose production in the liver. The expression of PGC1α strongly induces the gluconeogenesis genes PEPCK and G6P and promotes gluconeogenesis in the liver [35]. PGC1α is also regulated by its upstream factor AMPK, which can selectively inhibit liver gluconeogenesis, promote liver mitochondrial function, accelerate liver glucose catabolism, and promote systemic glucose homeostasis [36]. The genes and proteins related to lipid synthesis and gluconeogenesis in liver were analyzed by RT-PCR and WB. Cilostazol may inhibit lipid de novo synthesis and gluconeogenesis by activating the dual regulation of the AMPK-ACC1/SCD1 pathway and AMPK-PGC1α-G6P/PEPCK pathway, reduce lipid synthesis and glucose production in the liver, and correct glucose and lipid metabolism disorders in NAFLD mice under a high-fat diet, thereby improving liver lipid accumulation.

The abnormal composition of colonic microbial structure in mice fed a high-fat diet can increase intestinal permeability, resulting in increased bacterial penetration, decreased overall thickness of the mucus layer, the redistribution of tight junction proteins, and intestinal inflammation [37,38]. When the integrity of the tight junction protein is damaged, the intestinal permeability increases, which can lead to bacterial paracellular translocation, increase LPS in the blood, cause endotoxemia and subsequent liver inflammation, disrupt the regulation of glucose and insulin metabolism, and lead to NAFLD [39]. Cilostazol significantly reduced endotoxin levels in NAFLD mice. The pathological sections of colon tissue showed that Cilostazol administration caused the significant accumulation of inflammatory cells and a decrease in goblet cells in colon tissue of the NAFLD mice. Cilostazol also increased the expression of Claudin-1, Occludin, and ZO-1 related to tight junctions in the colon tissue and improved the intestinal structural integrity of the NAFLD mice, thereby avoiding intestinal microorganisms and their harmful products entering the hepatic portal vein to cause liver disease.

After proving that Cilostazol can protect the intestinal barrier of the NAFLD mice, this study continued to analyze the intestinal flora of mice using 16S rRNA technology. After Cilostazol intervention, the decrease in intestinal flora diversity in the NAFLD mice was improved, and the abundance of flora was restored. The structure of the intestinal flora was different from that of the other two groups. Compared with the Model group, Cilostazol intervention (Cilostazol-H group) increased the relative abundance of beneficial bacteria, such as Verrucomicrobiota, Bacteroidota, Akkermansiaceae, Oscillospiraceae, Bacteroidaceae, Akkermansia, and Lachnospiraceae _ NK4A136 _ group, related to reducing steatosis [40,41,42,43], anti-inflammation, and SCFA production and reduced the relative abundance of Desulfobacterota, Lactobacillaceae, Desulfovibrionaceae, Blautia, Desulfovibrioa, and Faecalibaculum related to promoting inflammation and metabolic disorders [44,45,46,47,48,49]. At the same time, after Cilostazol administration intervention, the Firmicutes/Bacteroidetes value was close to that of the Control group, which could preliminarily determine that Cilostazol could regulate the lipid metabolism of the host by intervening the intestinal flora, and its lipid-lowering mechanism may be related to the reduction in the F/B ratio. It is speculated that this inhibits the entry of pro-inflammatory molecules into the portal vein to the liver by improving the intestinal flora structure of NAFLD mice and enhancing the intestinal barrier, thereby reducing hepatocyte fat accumulation and hepatocyte damage [50]. LEfSe was used to analyze the differential species of intestinal flora in the mice. It was found that the iconic species of the Model group on the HFD diet were mainly g_Blautias_and s_Desulfovibrio_fairfieldensis, and the iconic species of the Cilostazol administration group were s_Akkermansia_muciniphila and f_Akkermansiaceae. The differential metabolites in the Cilostazol group were mainly related to the regulation of lipid metabolism and the improvement of intestinal inflammation, which was also confirmed in the correlation analysis between the intestinal bacteria and NAFLD-related indicators. Desulfovibrio was stably enriched in the NAFLD mice. Desulfovibrio was positively correlated with the lipid-related indicators that promote NAFLD and negatively correlated with the anti-inflammatory factor parameters. Cilostazol intervention significantly increased the quantity of Akkermansia and decreased the quantity of Desulfovibrio. Therefore, it is speculated that Desulfovibrio and Akkermansia are intestinal bacteria specifically regulated by Cilostazol and play a role in the treatment of NAFLD through the interaction of marker species. In the functional prediction of Tax4Fun, Cilostazol could significantly affect the expression of functional genes related to carbohydrate metabolism, amino acid metabolism, and energy metabolism in the intestinal microbial community of mice.

Intestinal microbial metabolite SCFAs are closely related to the hosts’ health and disease. This study found that after Cilostazol administration, the trend of SCFA levels in the NAFLD mice was reversed, and the SCFA levels were close to those in the Control group. The level and proportion of SCFAs are determined by the structural proportion of intestinal flora. If the proportion is imbalanced, it will lead to the occurrence and development of various diseases [51]. The correlation analysis between the intestinal bacteria and SCFAs showed that the composition of intestinal flora affected the level of SCFAs in the cecal contents. Because SCFAs can activate the AMPK signaling pathway to regulate the body’s lipid balance [19], it is speculated that Cilostazol promotes the production of SCFAs in the intestine by changing the abundance and structure of the intestinal flora of NAFLD mice, and then regulates its downstream signaling pathway to reduce the lipid synthesis process, and finally improves the liver lipid accumulation in NAFLD mice.

Although Cilostazol has a positive effect on the treatment of NAFLD, considering the individual differences in its bioavailability, it is still necessary to analyze its pharmacokinetics and determine active metabolites in the future to clarify the pharmacological effects of Cilostazol. Through correlation analysis, it was found that the intestinal flora affected the NAFLD indicators, which could be further verified by fecal bacteria transplantation experiments in the future. In summary, Cilostazol can improve glucose metabolism, the inflammation level, dyslipidemia, and the liver pathological status in NAFLD mice. The possible mechanisms of Cilostazol include activating the AMPK-ACC1/SCD1 and AMPK-PGC1α-G6P/PEPCK pathways, regulating the structure and abundance of intestinal flora, SCFAs, and the intestinal barrier function, and finally achieving the effect of treating nonalcoholic fatty liver disease.

## 4. Materials and Methods

### 4.1. Materials

Metformin was purchased from Sino-US Shanghai Squibb Pharmaceutical Co., Ltd. (Shanghai, China), batch number LTA007461EX2023/02 (Shanghai, China). Cilostazol was purchased from Zhejiang Dazhong Pharmaceutical Co., Ltd. (Zhejiang, China), batch number 201002P (Zhejiang, China). Specific pathogen-free 7-week-old C57BL/6J male mice were obtained from Hunan Silaike Jingda Experimental Animal Co., Ltd (Hunan, China). Common feed was purchased from Chengdu Hualanxu Biotechnology Co., Ltd (Chengdu, China). (carbohydrate 58.9%, protein 18.5%, crude fat 11.85%, fiber 3.2%, minerals 0.3%, and vitamin 1%). High-fat diet was purchased from Research Diets, Lot: D12492 (carbohydrate 26.3%, protein 26.2%, crude fat 34.9%, fiber 5%, mineral 3.5%, and vitamin 1%). Among the antibodies used in Western blotting, p-AMPKα was purchased from CST; the rest were purchased from Protein tech, including AMPKα, ACC1, p-ACC1, SCD1, PGC1α, G6P, PEPCK, Claudin-1, Occludin, ZO-1, and β-actin.

### 4.2. Experimental Design

Forty-five 7-week-old C57BL/6J male mice weighing 20–25 g were adaptively fed for 7 days in a specific pathogen-free experimental environment. The experiment met the ethical requirements of experimental animals in Guangdong Pharmaceutical University (gdpulacspf2017558). Subsequently, 8 mice were randomly selected as the blank group and fed a normal diet for 8 weeks. The remaining mice were fed a high-fat diet for 8 weeks as the Model group. One mouse was randomly selected from the blank group and two mice were randomly selected from the Model group for NAFLD-related blood lipid indicators and section detection. After judging the success of the nonalcoholic fatty liver model, the blank group was set as the Control group, with a total of 7 mice. The Model mice fed the high-fat diet were randomly divided into Model group, Metformin group (positive drug group, 50 mg/kg/d), Cilostazol-L group (Cilostazol low-dose group, 30 mg/kg/d), Cilostazol-M group (Cilostazol medium-dose group, 60 mg/kg/d), and Cilostazol-H group (Cilostazol high-dose group, 120 mg/kg/d), with 7 mice in each group. The recommended human dosage in the Cilostazol drug instructions was used as the low dose for the mice, and the medium-dose group and the high-dose group were set up with a 2-fold coefficient. According to the 0.1 mL/10 g gavage requirements of mice, the mice in each group were given the corresponding volume of sample solution for 12 weeks, and the mice in the Control group and the Model group were given the corresponding volume of distilled water.

During the period of gavage, the mice in each group were weighed and recorded regularly weekly. The oral glucose tolerance of mice was measured before the end of the experiment. After 12 weeks of intragastric administration of Cilostazol, the mice were fasted for 16 h, and then the sera and related tissues of the mice were sampled, including serum, liver, colon, and cecal contents. The above tissue samples were frozen in liquid nitrogen, and the treated sera were stored in an ultra-low temperature refrigerator.

### 4.3. Determination of Biochemical Parameters of Serum and Liver Tissue

The blood of mice was centrifuged at 3000 r/min to obtain sera. The TG, TC, HDL-C, LDL-C, FFA, ALT, and AST of serum samples were measured using common kits (Nanjing Jiancheng Bioengineering Institute, Nanjing, China). The levels of IL-6, IL-1β, IL-10, TNF-α, fasting insulin, fasting blood glucose, and LPS were measured using an ELISA kit (Jiangsu Feiya Biotechnology Co., Ltd., Jiangsu, China). The supernatant was obtained by centrifugation at 2500 r/min after homogenate of 0.1 g mouse liver tissue. TG, TC, HDL-C, and LDL-C were determined using a common kit (Nanjing Jiancheng Bioengineering Institute).

### 4.4. Histopathological Analysis

Appropriate amounts of liver tissue and intestinal tissue were dehydrated and embedded in paraffin, followed by sectioning and HE staining. An appropriate amount of liver tissue was dehydrated and embedded with OCT embedding agent for sectioning and oil red O staining. The stained slices were observed under a microscope (Olympus, Tokyo, Japan).

### 4.5. Real-Time PCR Analysis

Total RNA was extracted from liver tissues and colons using RNAiso Plus, and RNA was reverse-transcribed into cDNA using a reverse transcription kit (Takara Bio, Shiga, Japan). The related primer sequence is shown in Table 1. A fluorescence quantitative polymerase chain reaction was performed using the PCR amplification kit TB Green Premix Ex Taq^TM^ (Takara Bio, Japan) in a Roche PCR instrument. The amplification procedure was maintained at 95 °C for 8 s and performed at 60 °C for 40 cycles at 30 s/cycle. The relative expression of related genes was calculated by the 2^−ΔΔCt^ method with β-actin as the internal reference gene.

### 4.6. Western Blotting Analysis

RIPA and PMSF (Meilun Bio, China) were mixed at a ratio of 100:1 and added to the liver and colon tissues. The protein was extracted by centrifugation at 12,000 rpm, and the protein concentration was determined using the BCA kit (Biyuntian Bio, Beijing, China). Protein samples were obtained after metal bath denaturation at 98 °C. The samples were subjected to electrophoresis, membrane transfer, and antibody application. The final band was imaged in a developer (Bio-Rad, Hercules, CA, USA) using an ECL kit (Meilun Bio, Nanning, China), and the gray value of the band was analyzed using image J.

### 4.7. 16S rRNA Sequencing

The cecal contents of mice in the Control group, Model group, and CTZ group (Cilostazol-H group) were centrifuged at 1000 rpm for 10 min to obtain precipitates. The total DNA of the study samples was obtained using a DNA extraction kit, and PCR amplification was performed using 16S V4 region-specific primers (Phusion^®^ High-Fidelity PCR Master Mix with GC Buffer, Waltham, MA, USA). Primer sequence information is shown in Table 2. The obtained PCR products were subjected to gel electrophoresis and magnetic bead purification. Finally, library construction (TruSeq^®^ DNA PCR-Free Sample Preparation Kit, Waltham, MA, USA) and sequencing (NovaSeq6000, San Diego, CA, USA) were conducted.

### 4.8. SCFAs Analysis

The supernatant obtained by the centrifugation of 20 mg samples in the Control group, Model group, and CTZ group was added with MTBE solvent containing internal standard, and then centrifuged at 12,000 r/min. The supernatant was analyzed by GC-MS/MS (Agilent Tech, Santa Clara, CA, USA). A DB-FFAP (30 m × 0.25 mm × 0.25 μm) column was used with helium as the carrier gas. The injection volume was 2 μL, the column flow rate was 1.2 mL/min, and the solvent delay time was 3 min. After temperature programming, all the samples were analyzed by dynamic multiple reaction monitoring.

### 4.9. Data Statistical Analysis

The experimental data were analyzed by GraphPad Prism 9. T-test was used for comparison between the two groups, and one-way ANOVA was used for comparison between the two groups. The experimental data are presented as mean ± SD. The 16S rRNA data were analyzed on the online platform of the Metware Cloud platform (https://cloud.metware.cn/, accessed on 11 October 2022). Using the programming R language software (V4.0.0), Spearman correlation visualization was used to analyze the correlation between the intestinal flora and NAFLD-related indicators and the correlation between the intestinal flora and SCFAs. *p* < 0.05 was considered statistically significant. 

## Figures and Tables

**Figure 1 ijms-25-06280-f001:**
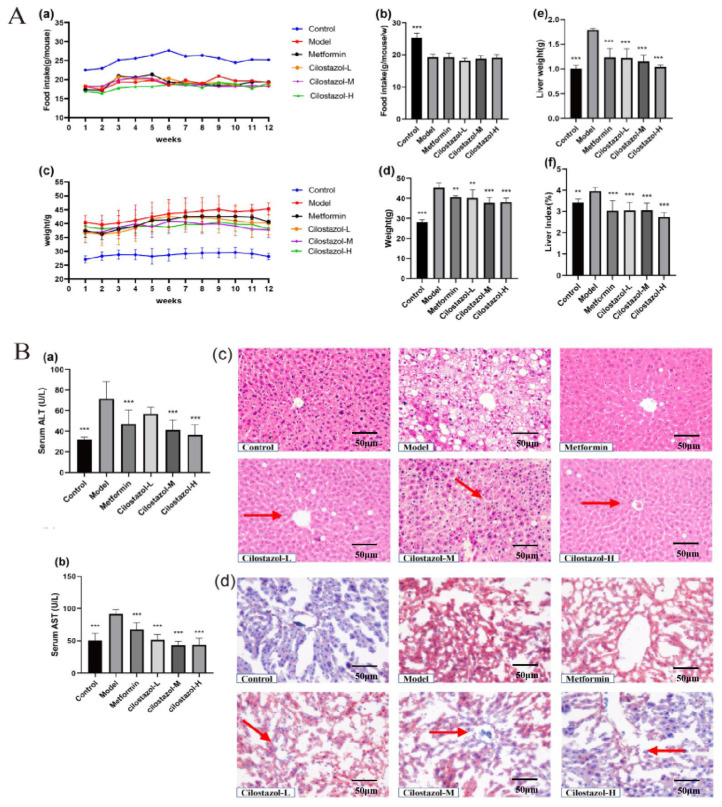
(**A**) The average food intake of mice during the treatment period (**a**); the average weekly food intake (histogram) (**b**); the weekly weight change in the mice (**c**); the weight of the mice at the 12th week (**d**); and, after treatment, the liver weight (**e**) and liver index (**f**) were measured. (**B**) The levels of ALT (**a**) and AST (**b**) in the sera of the mice after treatment (*n* = 7, mean ± SD). The HE staining of mouse liver tissue (**c**); the oil red O staining of mouse liver tissue (**d**). ** *p* < 0.01, *** *p* < 0.001 compared to the model group.

**Figure 2 ijms-25-06280-f002:**
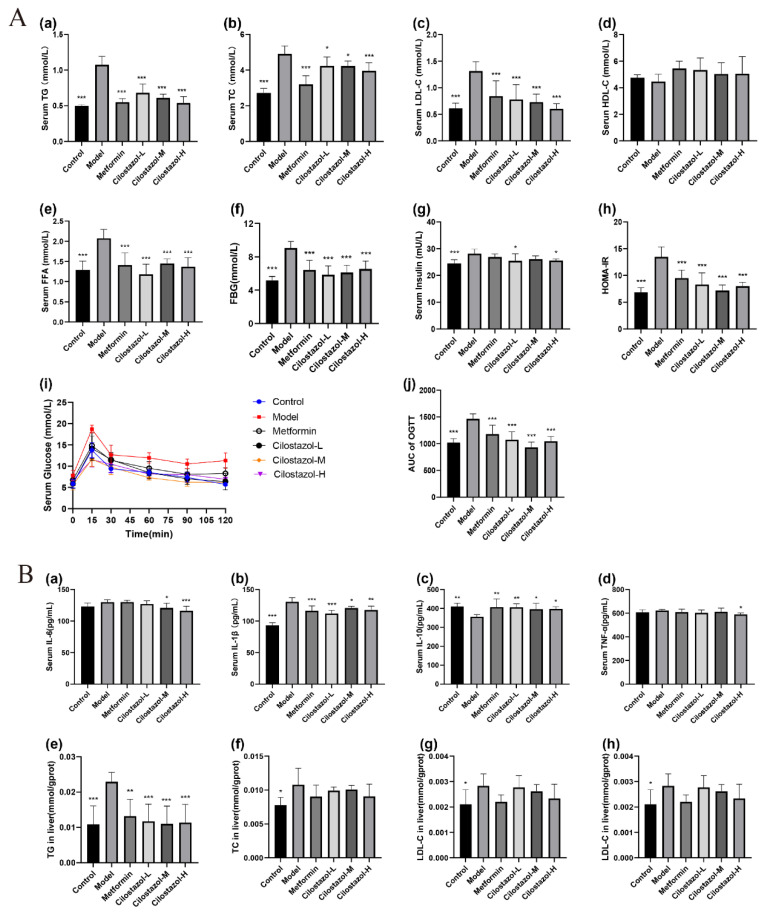
Blood serum lipid levels in mice after treatment (**A**). (**a**) TG, (**b**) TC, (**c**) LDL-C, (**d**) HDL-C, (**e**) FFA, (**f**) fasting blood glucose level, (**g**) fasting insulin level, (**h**) HOMA-IR index, (**i**) oral glucose tolerance change, and (**j**) area under the curve. Inflammation level and liver lipid content (**B**) in mice after treatment. (**a**) IL-6, (**b**) IL-1β, (**c**) IL-10, (**d**) TNF-α, (**e**) liver TG, (**f**) liver TC, (**g**) liver LDL-C, and (**h**) liver HDL-C (*n* = 7, mean ± SD). * *p* < 0.05, ** *p* < 0.01, *** *p* < 0.001 compared to the model group.

**Figure 3 ijms-25-06280-f003:**
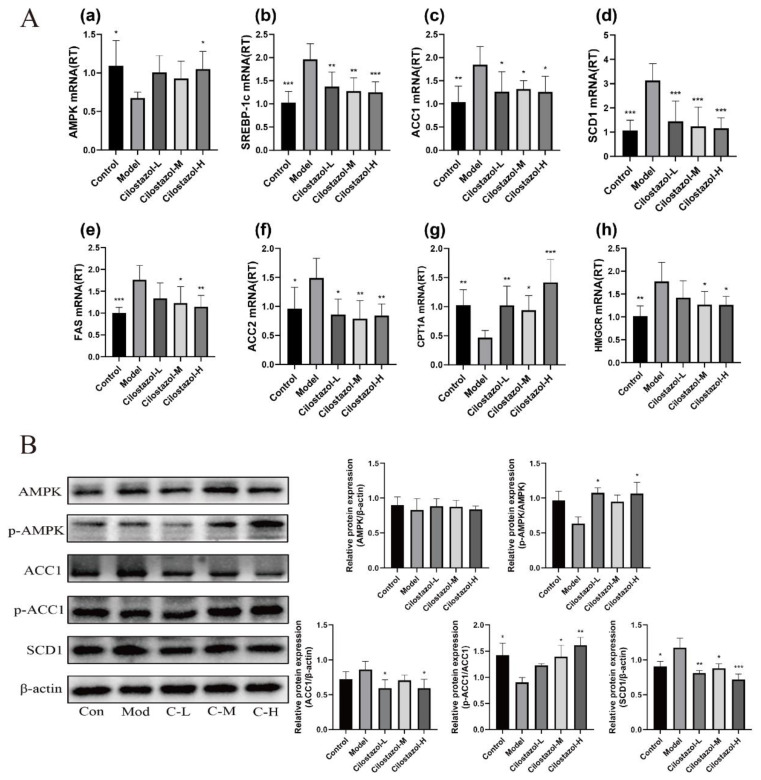
The relative expression of liver genes in each group (**A**) (*n* = 6, mean ± SD). AMPK (**a**), SREBP-1c (**b**), ACC1 (**c**), SCD1 (**d**), FAS (**e**), ACC2 (**f**), CPT1A (**g**), and HMGCR (**h**). The relative expression of liver protein in each group (**B**) (*n* = 3, mean ± SD). * *p* < 0.05, ** *p* < 0.01, *** *p* < 0.001 compared to the model group.

**Figure 4 ijms-25-06280-f004:**
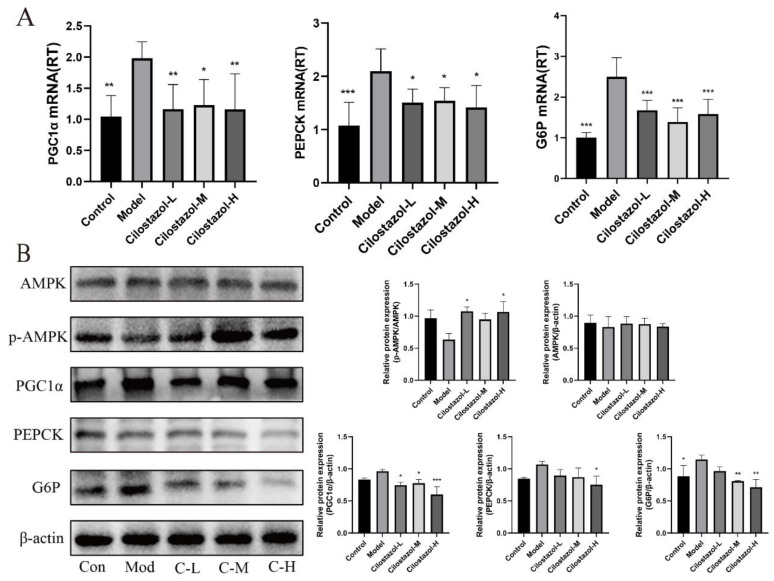
The gene expression of PGC1α, PEPCK, and G6P in liver tissues of each group (**A**) (*n* = 6, mean ± SD). The protein expression of AMPK, p-AMPK, PGC1α, PEPCK, and G6 P in liver tissues of mice in each group (**B**) (*n* = 3, mean ± SD). * *p* < 0.05, ** *p* < 0.01, *** *p* < 0.001 compared to the model group.

**Figure 5 ijms-25-06280-f005:**
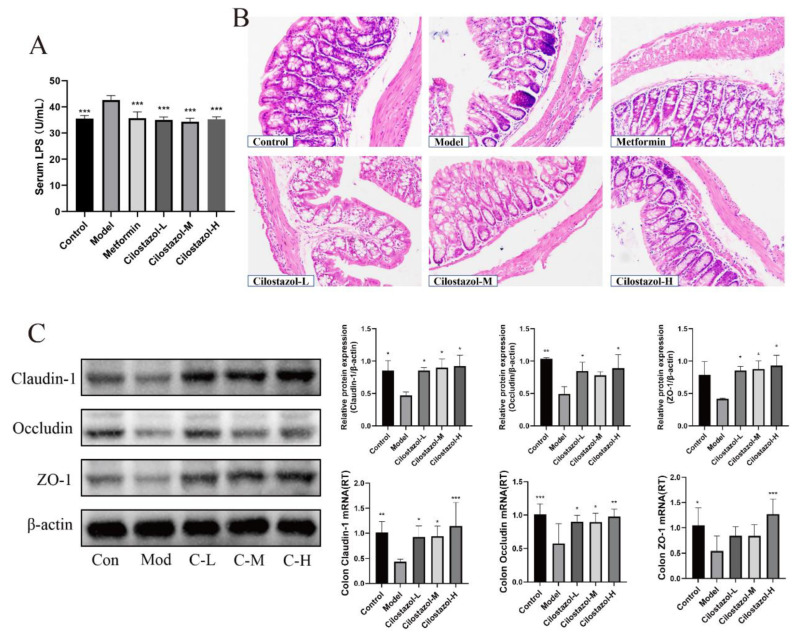
Serum LPS level (**A**) (*n* = 7, mean ± SD). HE staining of mouse colon tissue (**B**) (200×). The expression of Claudin-1, Occludin, and ZO-1 (**C**) genes (*n* = 6, mean ± SD) and proteins (*n* = 3, mean ± SD) in mouse colon. * *p* < 0.05, ** *p* < 0.01, *** *p* < 0.001 compared to the model group.

**Figure 6 ijms-25-06280-f006:**
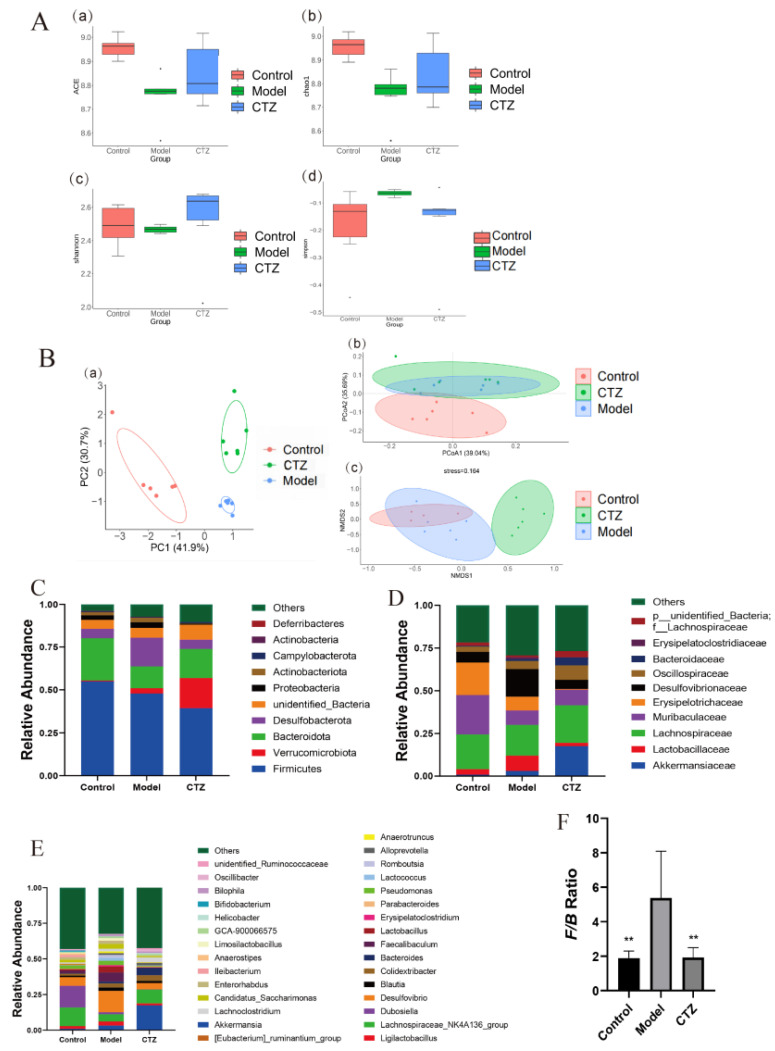
Alpha diversity analysis of intestinal flora in mice (**A**). (**a**) ACE index, (**b**) Chao1 index, (**c**) Shannon index, and (**d**) Simpson index. β diversity of intestinal flora in mice (**B**). (**a**) PCA analysis, (**b**) PCoA analysis, and (**c**) NMDS analysis. The community structure of intestinal flora in mice at the phylum level (**C**). the community structure of intestinal flora in mice at the family level (**D**). The community structure of intestinal flora at the genus level (**E**) (*n* = 6, mean ± SD). Firmicutes/Bacteroidetes ratio (**F**) (*n* = 4–5, mean ± SD). ** *p* < 0.01 compared to the model group.

**Figure 7 ijms-25-06280-f007:**
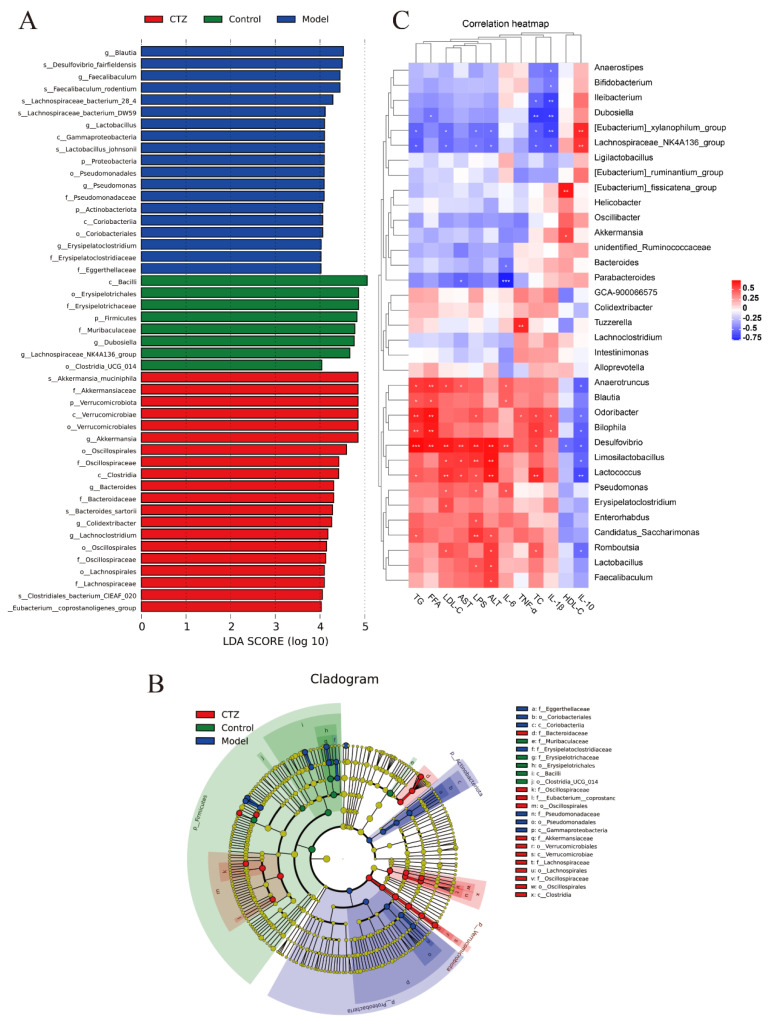
LDA analysis histogram of intestinal flora in mice (**A**); evolutionary branch diagram of mouse intestinal flora (**B**); correlation analysis between intestinal bacteria and NAFLD indexes (**C**). * *p* < 0.05, ** *p* < 0.01, *** *p* < 0.001 compared to the model group.

**Figure 8 ijms-25-06280-f008:**
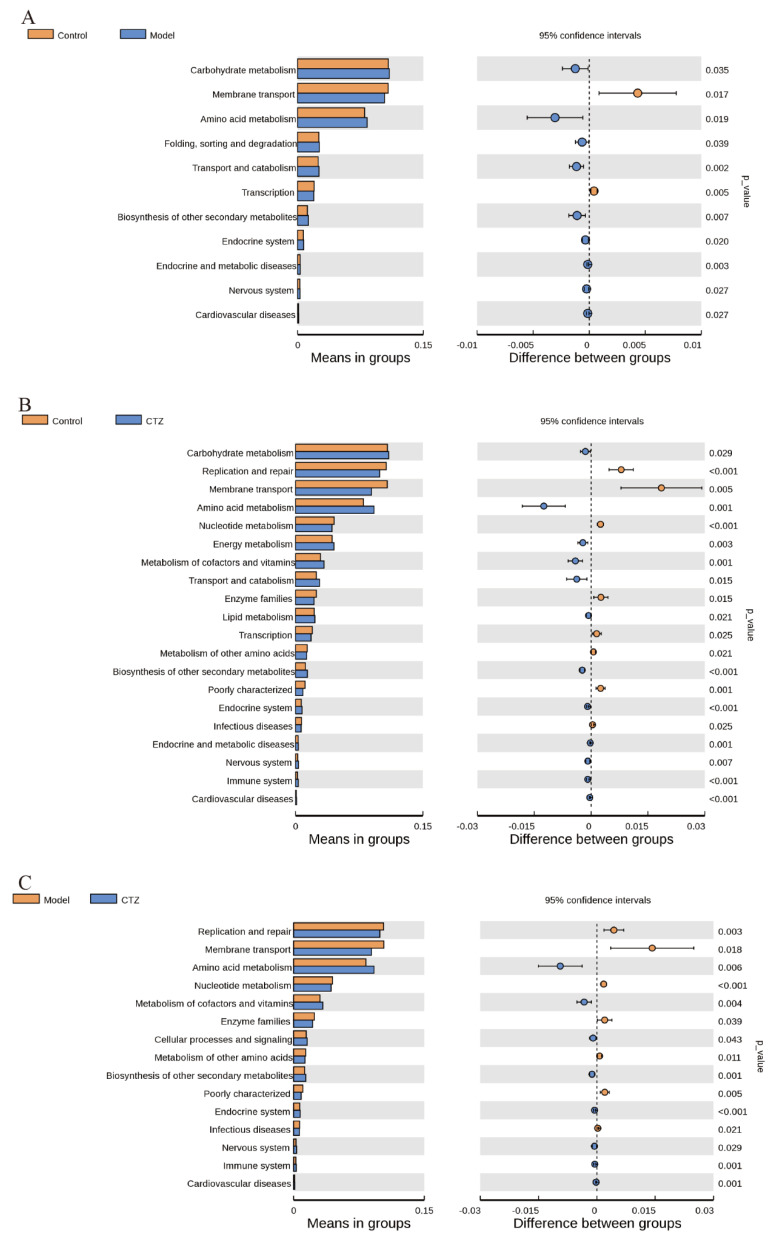
Functional prediction of intestinal flora in mice (level 2).The functional genes of intestinal flora in Control group and Model group (**A**);The functional genes of intestinal flora in Control group and Cilostazol group (**B**);The functional genes of intestinal flora in Model group and Cilostazol group (**C**).

**Figure 9 ijms-25-06280-f009:**
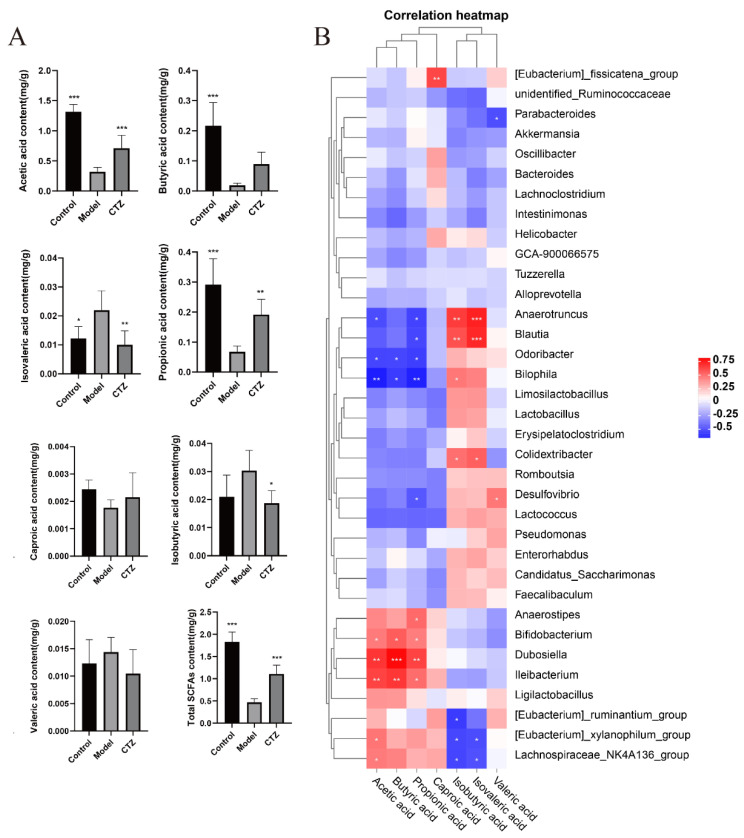
The content of short-chain fatty acids in the intestine of mice (**A**) (*n* = 6, mean ± SD); correlation analysis between intestinal flora and its metabolite short-chain fatty acids (**B**). * *p* < 0.05, ** *p* < 0.01, *** *p* < 0.001 compared to the model group.

**Table 1 ijms-25-06280-t001:** Primer sequences.

Gene	Forward Primer	Reverse Primer
*AMPK*	GGGAAAGTGAAGGTGGGCAA	GATGTGAGGGTGCCTGAACA
*SREBP-1c*	GGAGCCATGGATTGCACATT	GGCCCGGGAAGTCACTGT
*FAS*	GGGAAGCCCATCACCATCTTC	AGAGGGGCCATCCACAGTCT
*ACC1*	CCGCCAGCCTGAGTTCTTTT	ATCGGGAGTGCTGGTTTAGC
*SCD1*	TTCTTACACGACCACCACCA	CAGCCGAGCCTTGTAAGTTC
*ACC2*	GGCCAGTGCTATGCTGAGAT	AGGGTCAAGTGCTGCTCCA
*CPT1A*	TCGGTGAGCCTGGCCT	TTGAGTGGTGACCGAGTCTG
*HMGCR*	TGCCTGGATGGGAAGGAGTA	GCACCTCCACCAAGGCTTAT
*PGC1α*	GTCGGAAGACACCTTCCTCTC	AGAGCAGCACACTGGTTGG
*G6P*	CCAACGTATGGATTCCGGTGT	GCAAGGTAGATCCGGGACAG
*PEPCK*	GGGTGGAAGGTCGAATGTGT	AGCCCTTAAGTTGCCTTGGG
*β-actin*	CACTGTCGAGTCGCGTCC	TCATCCATGGCGAACTGGTG
*Claudin*	TGGGGCTGATCGCAATCTTT	CACTAATGTCGCCAGACCTGA
*Occludin*	GCCCCTCTTTCCTTAGGCG	TCCCAAGATAAGCGAACCTGC
*ZO-1*	AACCCCCATGGTGCTACTTC	CCTCCATTGCTGTGCTAGTGA
*β-actin*	CACTGTCGAGTCGCGTCC	TCATCCATGGCGAACTGGTG

**Table 2 ijms-25-06280-t002:** Primer information.

Sequencing Region	Primer Name	Sequence Information
V3 + V4	806R	GGACTACHVGGGTWTCTA
	515F	GTGCCAGCMGCCGCGGTAA

## Data Availability

Data contained within the article.

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
