# Peer review of "Cilostazol Attenuates Hepatic Steatosis and Intestinal Disorders in Nonalcoholic Fatty Liver Disease"

_ijms, 2024, doi:10.3390/ijms25116280_

Round 1
Reviewer 1 Report
Comments and Suggestions for Authors
This manuscript entitled "Cilostazol attenuates hepatic steatosis and intestinal disorders in nonalcoholic liver disease" is well-written and very important because NAFLD or MASLD pathogenesis, progression, and therapeutic strategy is still not clear enough. The authors aimed that examine the potential therapeutic effects of Cilostazol on lipid and glucose metabolism, as well as on intestinal flora in NAFLD mice and investigate the signaling putways.
Major revision:
Title: Instead ....nonalcoholic liver disease, put ....nonalcoholic fatty liver disease
Abstract: In first sentence: Nonalcoholic fatty liver disease (NAFLD) is one of the most common liver diseases in the world, which begins with liver lipid accumulation and is associated with metabolic syndrome, add ......the most common chronic liver diseases. Also, the name chosen to replace NAFLD was metabolic dysfunction–associated steatotic liver disease (MASLD) (add) (Rinella ME, Lazarus JV, Ratziu V, Francque SM, Sanyal AJ, Kanwal F, Romero D, Abdelmalek MF, Anstee QM, Arab JP, Arrese M, Bataller R, Beuers U, Boursier J, Bugianesi E, Byrne CD, Castro Narro GE, Chowdhury A, Cortez-Pinto H, Cryer DR, Cusi K, El-Kassas M, Klein S, Eskridge W, Fan J, Gawrieh S, Guy CD, Harrison SA, Kim SU, Koot BG, Korenjak M, Kowdley KV, Lacaille F, Loomba R, Mitchell-Thain R, Morgan TR, Powell EE, Roden M, Romero-Gómez M, Silva M, Singh SP, Sookoian SC, Spearman CW, Tiniakos D, Valenti L, Vos MB, Wong VW, Xanthakos S, Yilmaz Y, Younossi Z, Hobbs A, Villota-Rivas M, Newsome PN; NAFLD Nomenclature consensus group. A multisociety Delphi consensus statement on new fatty liver disease nomenclature. Hepatology. 2023 Dec 1;78(6):1966-1986. doi: 10.1097/HEP.0000000000000520. Epub 2023 Jun 24. PMID: 37363821; PMCID: PMC10653297.
Introduction: NAFLD is often associated with obesity, diabetes, hypertension, and hyperlipidemia. ..Since it is associated with these comorbidities, it was renamed to MASLD (ref. A multisociety Delphi consensus statement on new fatty liver disease nomenclature. Hepatology. 2023 Dec 1;78(6):1966-1986. doi: 10.1097/HEP.0000000000000520.)
The Multiple-Hit pathogenesis of NAFLD write in more detail. In NAFLD, lipids initially accumulate in hepatocytes and develop into reversible hepatic steatosis followed by infiltration of immune cells in the liver, consequently increasing the inflammatory process and triggering liver fibrosis.[8]
Paragraph 4: Cilostazol is a 2-hydroxyquinolone derivative and a drug approved for improving claudication distance (add references). Studies have shown that Cilostazol can ameliorate hepatic steatosis (add references), but the specific mechanism is still unknown.
Paragraph 5: In this study, a mouse model of NAFLD was induced by high fat diet (HFD),the mechanism of Cilostazol in the treatment of NAFLD was studied based on the AMPK pathway, and the effect of Cilostazol on the intestine of NAFLD mice was ulteriorly discussed. (reformulate, that is, write better).
2. Results
2.1. Cilostazol reduced body weight and liver weight in NAFLD mice
In Figure 1 (Bc and Bd): mark the main histopathological changes of liver tissue (stars, arrows...)
2.2. Cilostazol improved liver function and reduced lipid accumulation in NAFLD mice.
In this Paragraph delete next sentence: HE staining and oil red O stain
ing were performed on liver paraffin sections of mice in each group.
Figure 2. Blood Serum lipid levels ....
In Figure 5B mark the expression of Claudin-1, Occludin and ZO-1
4.2. Experimental design
Why did the authors decide on a 12-week treatment with Cilostazol?
Second Paragraph: .....After 12 weeks of intragastric administration of Cilostazol,...
4.4. Pathological observation Histopathological analysis
Minor revision:
Check all manuscript for grammar errors ( for example: first the reference is stated, followed point)
Reviewer 2 Report
Comments and Suggestions for Authors
In this manuscript, Authors demonstrated a beneficial role of Cilostazol in a mice model of nonalcoholic liver disease (NAFLD), improving liver steatosis by the inhibition of liver lipid de novo synthesis and gluconeogenesis, by regulating AMPK-ACC1/SCD1 and AMPK-PGC1α-G6P/PEPCK pathways. They demonstrated also that Cilostazol ameliorated intestinal dysfunction by normalizing intestinal permeability and microbiota composition and increasing the level of short-chain fatty acids in NAFLD mice to a level similar to that in the control group. Authors propose Cilostazol as a new drug for treating NAFLD.
The paper is well written, overall clear and results well support the conclusions. Experimental design is appropriate. I have only some comments:
- It is not clear if during the 12 weeks of Cilostazol treatment mice continue to be fed with the fat diet or are subjected to a normal diet. From graphic Ac in Fig.1 their weight seems to further increase while I expected a light decrease; Authors should specify and comment this result;
- In the graphics showing alpha diversity analysis of microbiota the significance is missing;
- In experiments on microbiota analysis only one group treated with Cilostazol is present corresponding to the high concentration of CTZ. This should be specified in the description of experiments other than in material and method section;
- Page 2 lane 70 a reference is missing;
- Page 6 title of the section 2.5 the acronym DNL should be explained;
- For a better reading, the font size in the graphics should be larger, in particular in figure 6.
Round 2
Reviewer 1 Report
Comments and Suggestions for Authors
The authors accepted all suggestions and remarks and corrected the manuscript. My opinion is that this article is now suitable for publication.